# Experimental study of water migration characteristics in compacted loess subjected to rainfall infiltration

Shibin Zhang[1], Tielin Han[1]*, Yani Lu[2], Chengzhi Huang[3], Peng Zhao[3]

**1** State Key Laboratory Base of Eco-hydraulic Engineering in Arid Area, Xi'an University of Technology, Xi'an, China, **2** School of Civil Engineering, Hubei Engineering University, Xiaogan, China, **3** China Shipbuilding Industry Institute of Engineering Investigation and Design Co., Ltd., Shanghai, China

* s3050210133@163.com

**Data Availability Statement:** All the data used to support the findings of this study are included within the paper.

**Funding:** This research was funded by the National Natural Science Foundation of China (Grant No. 41907259) (Yani Lu) and Natural Science

## Abstract

In order to study the effect of the rainfall infiltration on water migration in compacted loess, a model device was developed for testing water migration in the soil under rainfall conditions. In this study, the volumetric water content and resistivity of soil were introduced into the model test device. This model test device was applied to the study of water migration characteristics in compacted loess under different rainfall conditions. The results show that the resistivity decreases with the increase of the volumetric water content at the same depth of the loess column. In this way, the characteristics of the water migration can also be reflected from the change of the resistivity. There is an intimate relationship between the resistivity and volumetric water content, dry density. The volumetric water content and dry density are normalized by saturation of loess, arriving the equation of saturation against the resistivity. The characteristics of rainfall infiltration in compacted loess show a particular pattern, which demonstrates that, with the increase of dry density of the loess column, the rainfall infiltration line present "Y", "D" and "Λ" shape distribution respectively, under light rain, heavy rain and rainstorm.

## Introduction

In Northwest China, loess is mostly unsaturated soil, in which the water sensitivity, permeability and unsaturated characteristics of unsaturated soil determine the strength and deformation characteristics of loess [1–3]. Under the condition of rainfall infiltration, water migrates in the form of unsaturated seepage [4]. Understanding the characteristics of water migration in compacted loess is crucial to studying the stability of loess slope under rainfall infiltration [5, 6] and the water migration in the loess foundation of high-filling engineering [7, 8].

The strength and deformation of soil are the most important mechanical properties for slope engineering and high-filling engineering under rainfall condition. By analyzing the stability of slopes under different rainfall intensities, Shimada et al. [9] concluded that the increase of saturation in the slope led to the reduction of matrix suction under rainfall, which had a great impact on the stability of the slope. Aristizabal et al. [10] studied the influence of

Foundation of Shaanxi Provincial of China (Grant No. 2021JQ-463) (Tielin Han), and the funders had role in study design, data collection, decision to publish.

**Competing interests:** The authors have declared that no competing interests exist.

rainfall intensity on the infiltration and deformation characteristics of unsaturated soil slope using the numerical analysis method, and concluded that the increase of pore water pressure (the reduction of suction) was mainly concentrated near the slope in the process of rainfall infiltration. Haeri et al. [11] studied the effects of slope geometry, unsaturated soil properties, groundwater level location and the infiltration on slope failure resistance and landslide stability. The study of water migration characteristics of soil under rainfall infiltration using the theory of unsaturated soil emerged late in China [12]. Based on Fredlund's theory on the strength of unsaturated soil [13], Li [14] analyzed the correlation between matrix suction and volumetric water content through the soil water characteristic curve measured in the test, obtaining the shear strength of soil, and the stability coefficient of slope. Taking an actual landslide in Chongqing as an example, Sun et al. [15] studied the influence of rainfall infiltration on the slope stability and the seepage field by using the finite element and limit equilibrium methods, and concluded that rainfall infiltration led to significant changes in the seepage field, especially when the pore water pressure of slope increased greatly, the deep slope failure was prone to occur. The stability analysis mechanism of unsaturated soil slope is complicated and greatly affected by various factors under rainfall infiltration. Liu et al. [16] deduced the best fitting curve between infiltration time, infiltration depth, infiltration rate and wetting front saturation of unsaturated soil slope under rainfall infiltration by analyzing the variation of initial water content of slope.

Based on the study of water migration in soil under different rainfall intensities, many researchers at home and abroad have done many related field immersion tests and artificial rainfall tests. Lee et al. [17] analyzed the meteorological data of a slope site and concluded that the main factor of the slope instability was the ratio of rainfall intensity to the saturated permeability of soil. Zhou et al. [18] studied the effects of water migration in slope on pore water pressure, safety factors and displacement of slope under rainfall, and concluded that the safety factor of slope was changing during rainfall infiltration. The wetting depth of slope is an important index to evaluate the slope instability induced by rainfall. Tu et al. [19] studied an expressway slope located in the loess plateau, using test devices such as a water sensor and pore water pressure gauge embedded in the highway slope, which can provide real-time monitoring, and they acquired the relationship between the downward movement of wetting front and the rainfall intensity. It was necessary to characterize the field infiltration and wet front movement induced by natural rainfall. Kim et al. [20] conducted field monitoring on a dense roadside slope of a highway in South Korea and studied the infiltration characteristics of the field slope by measuring the changes of matrix suction and water content. Zhang et al. [21, 22] studied the water migration characteristics in loess under the rainfall condition in the real-world setting by carrying out artificial rainfall test and measuring the depth of water infiltration using the water sensor. Sun et al. [23] carried out full-scale field slope rainfall tests under different rainfall conditions and studied the surface infiltration process and deformation characteristics of unsaturated loess slope, and the results showed that the water infiltration depth and infiltration rate increase with the increase of rainfall intensity during rainfall. In the above research, the monitoring time is limited, and the monitoring method is relatively single for the study on the water migration in soil. In addition, there is little information about the conclusions of water migration in compacted loess foundation of high-filling engineering under rainfall infiltration.

Therefore, a model test device was developed in this study, which could reflect the behaviour of water migration characteristics in soil under different rainfall conditions. The volumetric water content and conductivity of compacted loess were introduced into this model test device, and the laboratory model tests of water migration were carried out in compacted loess columns under different rainfall conditions, analyzing the effects of water migration on the

volumetric water content and resistivity of loess. Then the water migration characteristics in compacted loess columns were then studied from the meso-level perspectives of volumetric water content and resistivity of loess, which could provide a new methodology to study the stability of loess slope and the characteristics of water migration in compacted loess foundation of high-filling engineering under rainfall infiltration.

## Design of model test device of water migration characteristics

### Basic introduction of model test device

The equipment used in the model test in this study is a self-developed model test device for measuring water migration, and its schematic diagram is shown in Fig 1. Its basic introduction is as follows: a plexiglass cylinder is the most important part of the model test device, which has an outer diameter of 32 cm, an inner diameter of 30 cm and a height of 80 cm. The core part is two measurement systems, namely water measurement system and resistivity measurement system.

In the model test device for measuring water migration, the plexiglass cylinder can be divided into several layers from top to bottom, and three through holes are evenly opened in each layer along the diameter direction. The moisture sensor is placed in a through hole, which is connected to the multi-channel junction box through the wire, and then the junction box is connected to the environmental tester, to form the water measurement system of the model test device. A small rectangular copper electrode is pasted in the two holes along the

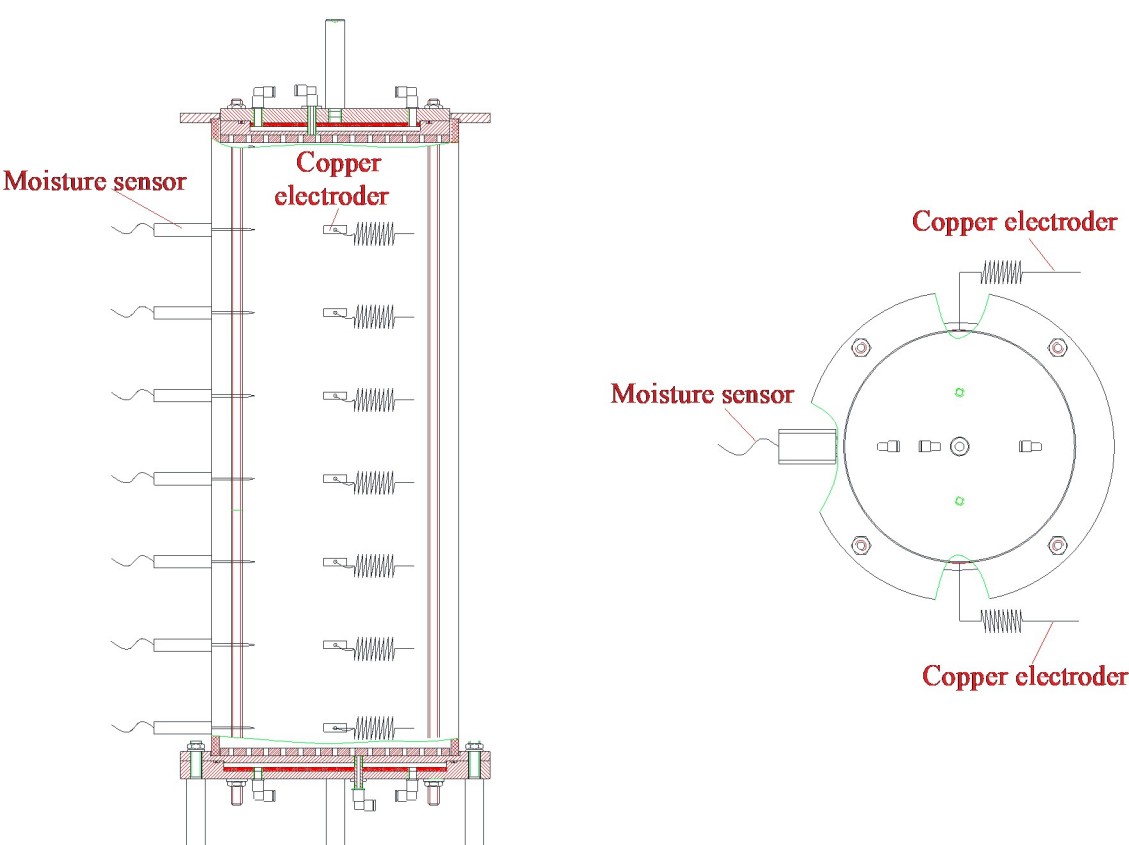

**Fig 1. Structure diagrams of model test device.** (a) Main view, (b) Vertical view.

diameter direction, and the pair of copper electrodes are connected to the resistance tester through wires, which constitutes the resistivity test system of the model test device. The model test equipment has the characteristics of convenient operation, simple structure and stable control.

## Working principle of model test device

The working principle of the model test device is that in the process of the measurement of water content, the moisture sensor is used to probe the change of water content of the model sample, then the change of water content is transmitted to the environmental tester through the multi-channel junction box, and the volumetric water content of soil is read by the environmental tester. The environmental tester can monitor and record automatically, the volumetric water content in the loess column can be measured in real time, to complete the process of moisture measurement.

In the process of resistance measurement, the resistance ($R$) of each layer along the diameter direction in the loess column is directly measured by the resistance tester, which can be converted into the resistivity ($\rho$) of the loess column using the expression ($\rho = RS/L$), where $L$ is the radial length of the loess column and $S$ is the area of the copper electrode. Based on the ingenious dimension of the copper sheet electrode with 3 cm in length, and 1 cm in width as well as 30 cm in radial length of the loess, and $R = 10^3\rho$ can be obtained. When the resistance measuring instrument is set at $10^3$ mode, the value displayed on the resistance measuring instrument is exactly the value of $\rho$.

Therefore, through the model test device of water migration characteristics developed in this study, the volumetric water content and resistivity of each layer of loess in the compacted soil column can be measured, so as to establish the relationship between loess resistivity and volumetric water content, revealing the law of water migration in compacted loess under different rainfall intensities.

## Operation steps of model test device

According to the characteristics of the developed model test device of water migration characteristics, the operation steps of the model test device are expressed as follows. The process of sample preparation is shown in Fig 2.

1. Inspect and calibrate the developed test device, including water measurement system and resistivity measurement system, so that it meets the test requirements.

2. Assembly of model test device: firstly, connect the lower bottom plate and lower cover plate with the base, install the plexiglass cylinder on the lower cover plate of the lower base, fix the four pull rods on the lower bottom plate and lower support plate, then put the upper support plate through the pull rod and cover the upper part of the plexiglass cylinder, and then fix the upper support plate and pull rod.

3. Preparation of test soil column: put the prepared loess samples into the plexiglass cylinder, fill in layers and compact layer by layer according to the compactness required for the test. After the preparation of the loess column is completed, put the perforated plate on the top of the loess column.

4. Measurement system installation: The moisture sensor is inserted into the through hole of each layer of the prepared soil column, and then the environmental tester is connected, which forms the water measurement system. The resistance measuring instrument is connected to a pair of radial electrodes, which constitutes the resistivity measurement system.

(a)

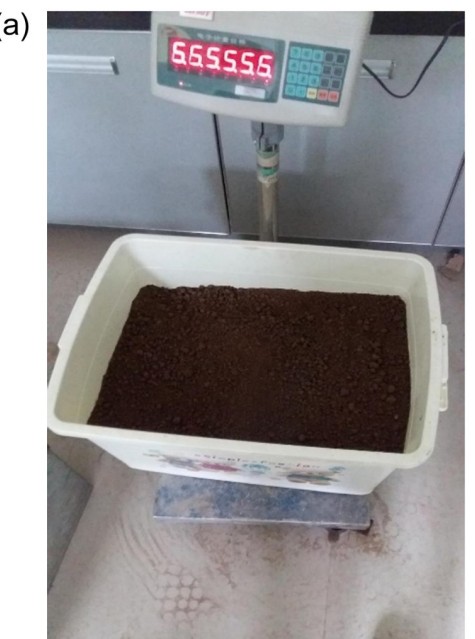

(b)

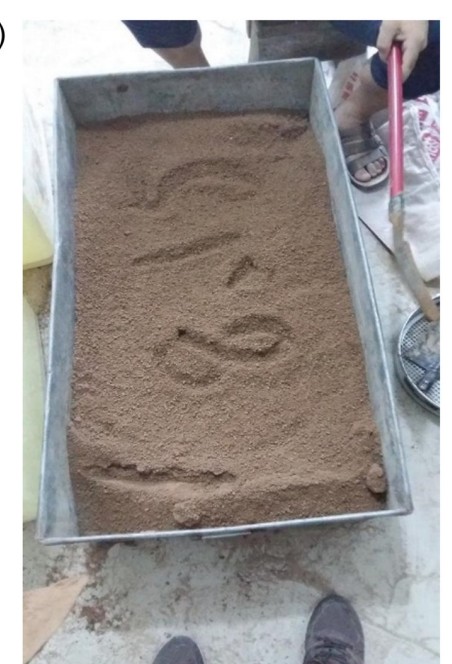

(c)

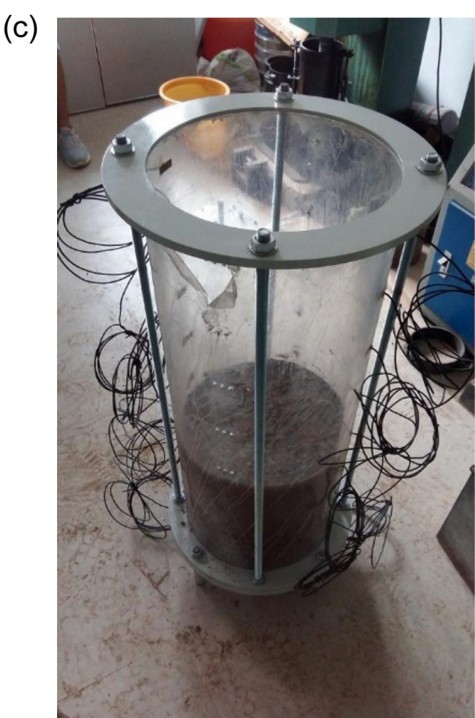

(d)

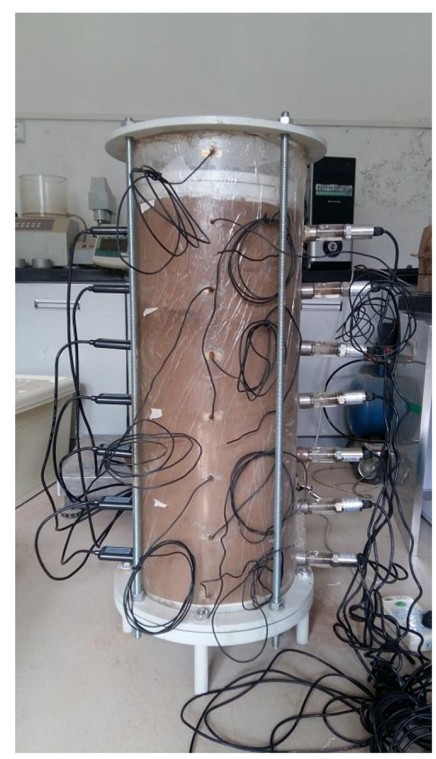

**Fig 2. Sample preparation processes.** (a) Weigh soil samples, (b) Preparation of soil samples, (c) Layered filling soil column, (d) Measurement system installation.

Through the above operations, a complete test device system is assembled. Note that it should be sealed with raw tape in this process.

5. Collection of test data: Tests are carried out according to the set testing schemes. During the test, the measurement of volumetric water content requires moisture sensor probing the change of water, and then transmitted to the environmental tester to complete the measurement process. For the measurement of resistivity, just turn on the resistance measuring instrument to directly measure the soil resistance at any time, and the measured soil resistance can be transformed into soil resistivity.

6. Remove each measurement system after the test. Take down the pressure bar, upper cover plate and perforated plate in turn, remove the four pull rods, take out the test soil sample from the plexiglass cylinder, and sort out the test device.

## Materials and methods

### Materials

The loess samples were taken from the construction site of loess slope engineering with a latitude and longitude of 36°38′ N, 109°32′ E in New District of Yan'an in China. The sampling depth was 2.5 m-3.5 m, at which the properties of loess were uniform. The main physical parameters of loess are shown in Table 1.

### Methods

The samples are remolded loess in this study, and their initial water content is the optimum water content $w_{op}$ = 16.0%. Three columns filled with compacted loess of different dry densities are prepared ($\rho_d$ = 1.42g/cm$^3$, 1.50g/cm$^3$, and 1.58g/cm$^3$), which is 30 cm in diameter and 70 cm in height. Three dry densities represent the loess columns with different compaction degree, and the increase of dry density in turn can reflect the gradual compaction of the loess columns. The loess column with dry density of 1.42 g/cm$^3$ is easy to be compacted, while it is difficult to be compacted at dry density of 1.58 g/cm$^3$ in the actual sample preparation process. Fig 3(a) shows the model test device of water migration characteristics. After the loess column is prepared, it is divided into four layers from top to bottom with the top of the loess column as the starting point (Z = 3.2 cm, 24.6 cm, 46.0 cm, and 67.4 cm), and the stratification of the loess column is shown in Fig 3(b).

Three different types of rainfall, namely light rain, heavy rain, and rainstorm, are artificially created at the top of the same loess column, and the duration of different rain types is calculated as 24 hours. Taking the standard rain type as a reference [24], that is, the daily rainfall of light rain in this model test is 7.1 mm (standard light rain: 0–10.0 mm), heavy rain is 35.4 mm (standard heavy rain: 25.0–49.9 mm), and rainstorm is 63.7 mm (standard rainstorm: 50.0–99.9 mm). In the process of model test of water migration characteristics, the resistivity measurement system and the water measurement system are used to measure the resistivity ($\rho$) and volumetric water content ($\theta_v$) at the loess column depths of Z = 3.2 cm, 24.6 cm, 46.0 cm, and 67.4 cm, respectively.

**Table 1. Main physical parameters of loess.**

| Material | Natural water content (%) | Optimum water content (%) | Maximum dry density (g/cm$^3$) | Soil specific gravity Gs | Liquid limit (%) | Plastic limit (%) | Plasticity index |
|---|---|---|---|---|---|---|---|
| Loess | 11.6 | 16.0 | 1.77 | 2.7 | 28.7 | 21.4 | 7.3 |

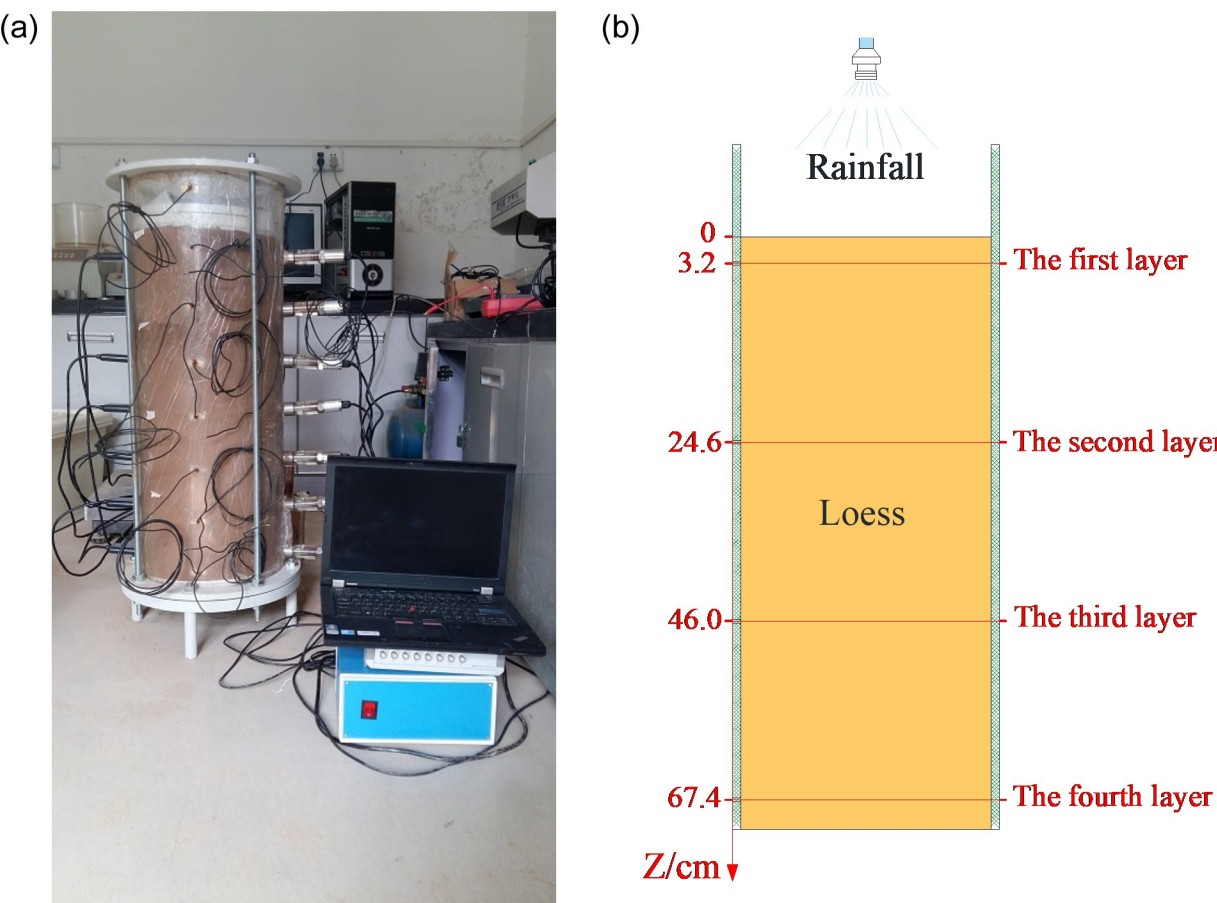

**Fig 3. Self-developed model test device of water migration characteristics.** (a) Model test device, (b) Layering of loess column.

The method of slowly spraying and adding water level by level is adopted to simulate light rain, heavy rain and rainstorm. The next level of rainfall was simulated until the previous level of rainfall has ended and reached a basically stable state (that is, the volumetric water content and resistivity values of each layer of soil in the compacted loess column are measured basically unchanged). The top of the plexiglass cylinder is sealed with plastic film to prevent water evaporation after each rainfall simulation.

## Results

### Analysis of variation law of volumetric water content with duration

The model tests were carried out to study the variation laws of the volumetric water content of loess with the duration of water migration in compacted loess of different dry densities under different types of rainfall. Fig 4 shows the relationship between the volumetric water content of the loess column and the duration of test at dry density of $\rho_d$ = 1.42 g/cm$^3$ and 1.58 g/cm$^3$.

Combined with Figs 3(b) and 4, it can be concluded that during the downward migration of water in the loess column, the depth $Z$ = 3.2 cm is located at the top layer of the loess column, where the volumetric water content of loess changes most significantly and the water migration rate is also the fastest. At the first layer and second layer of the loess column ($Z$ = 3.2 cm and 24.6 cm), under each rainfall intensity, with the continuous rainfall infiltration time,

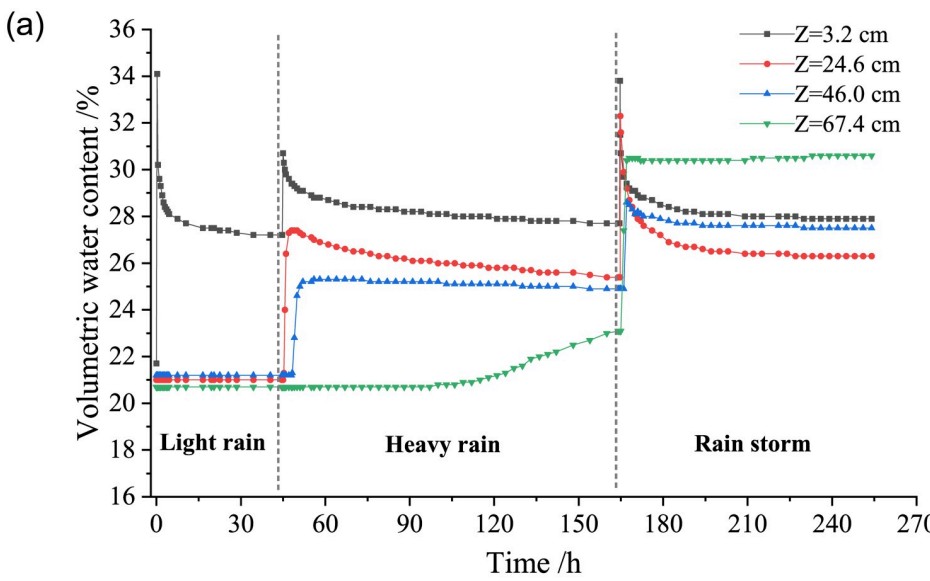

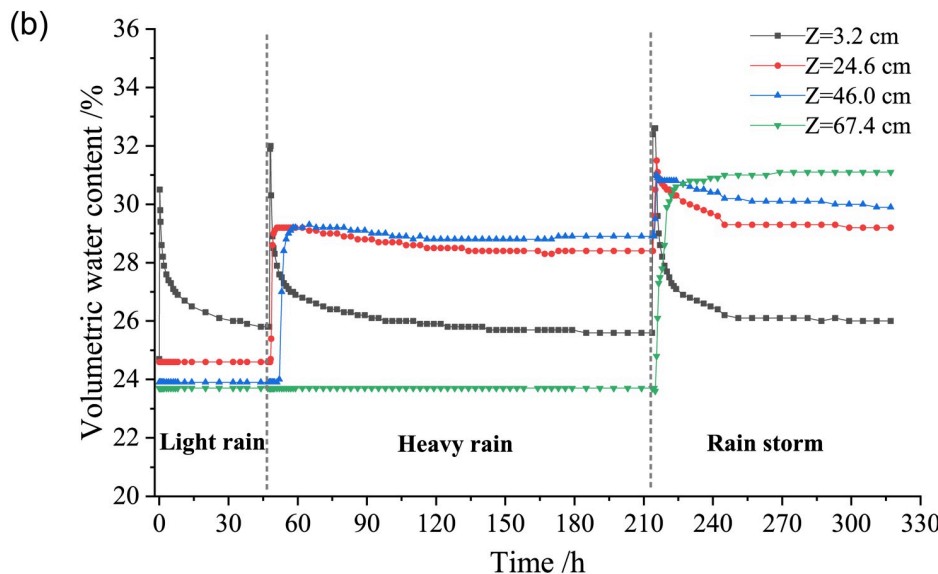

**Fig 4. Variation curves of volumetric water content of loess with duration under different dry densities.** (a) $\rho_d =$ 1.42 g/cm³, (b) $\rho_d =$ 1.58 g/cm³.

the volumetric water content of loess first rises sharply, and then decreases slowly to reach the basic stable state, which also shows that the water in the upper layer of the loess column gradually migrates to the lower layer because its own gravity and internal matrix suction of loess. In the stage of light rain and heavy rain, at the third layer of the soil column ($Z$ = 46.0 cm), the volumetric water content of loess first rises rapidly and then gradually tends to reach a basically stable state, reflecting the characteristics of water migration to the deep layer of soil. However, in the rainstorm stage, the volumetric water content of loess gradually increases with the duration of rainfall infiltration time, and finally reaches a basically stable state. At the fourth layer of the loess column ($Z$ = 64.7 cm), the volumetric water content of loess increases slowly at first, with the continuation of the test time, the depth of water migration increases under

rainstorm, then the volumetric water content at this depth rises sharply and finally tends to be basically stable.

In the stage of light rain and heavy rain, due to the limited total rainfall and the high dry density of the loess ($\rho_d$ = 1.58g/cm³), the water cannot migrate to the fourth layer (Z = 64.7 cm). However, in the stage of rainstorm, with the increase of total rainfall, the water gradually migrates to the depth of 64.7 cm, the volumetric water content of the loess column shows a rapid rise, and a small amount of water can be observed to seep from the bottom drain of the model test device, indicating that the water migration depth has exceeded 70 cm. The higher the dry density of the loess column is, the longer it takes for the water migration to reach the basic stable state in the loess column. The time it takes for the loess column with dry density of 1.58 g/cm³ to reach the basic stable state is significantly longer than that of other soil columns during water migration. When the dry density of the loess column is small, its compaction degree is low, and the porosity of soil is relatively large. Under the same amount of rainfall, the rainfall infiltration rate is faster. With the increase of dry density of loess column, the soil is gradually compacted and the porosity of soil decreases. The water infiltration rate decreases and the water transport capacity is poor. Therefore, the time required for the rainfall infiltration to reach the basic stable state increases significantly during water migration in the loess column.

## Analysis of variation law of resistivity with duration

Under different conditions of rainfall infiltration, the resistivity of loess changes with the water migration in compacted loess. To study the influence of the resistivity on the water migration characteristics in compacted loess, the relationship curve of the resistivity of loess with duration is shown in Fig 5.

It can be seen from Fig 5 that the migration of water to different depths of the loess column cause a sudden sharp decrease in the resistivity of the soil under different types of rainfall. The reason is that under the action of rainfall, the soil saturation gradually increases, the pores between soil particles are rapidly filled with water, and some ions in the soil dissolve into the pore water, which enhances the conductivity of soil, resulting in a sudden decrease in the resistivity of soil [25, 26]. During the downward migration of water, the resistivity of loess at the depth Z = 3.2 cm of the first layer of the loess column is most sensitive. The resistivity of loess decreases rapidly at the initial moment of rainfall, increases slowly with the test duration, and then gradually reaches the basic equilibrium state. And the change of resistivity at the depth of other layers of the soil column lags behind that of the first layer. At the initial moment of rainfall, the amount of water in the surface layer of the loess column increases sharply, and it is too late for the water to infiltrate into the deep layer of the loess column. The saturation of surface soil rapidly increases, which makes the resistivity greatly decrease. As the rain infiltrates into the deep layer of the loess column, the saturation of the surface soil gradually decreases until it reaches an equilibrium state, while the saturation of the deep soil gradually increases, and the resistivity of soil changes accordingly. Under each rainfall intensity, at the depth of soil column (Z = 3.2 cm and 24.6 cm), with the continuous rainfall infiltration, the resistivity of loess decreases sharply and then increases gradually to reach a basically stable state. This also reflects that as the rain infiltrates into the deep layer of the loess column, the water volume between soil pores increases rapidly at first and then decreases gradually until the water no longer migrates to the deeper layer, reaching a basic equilibrium state.

In the stage of light rain and heavy rain, at the third layer of the soil column (Z = 46.0 cm), with the duration of the test, the resistivity of loess first decreases rapidly, and then gradually

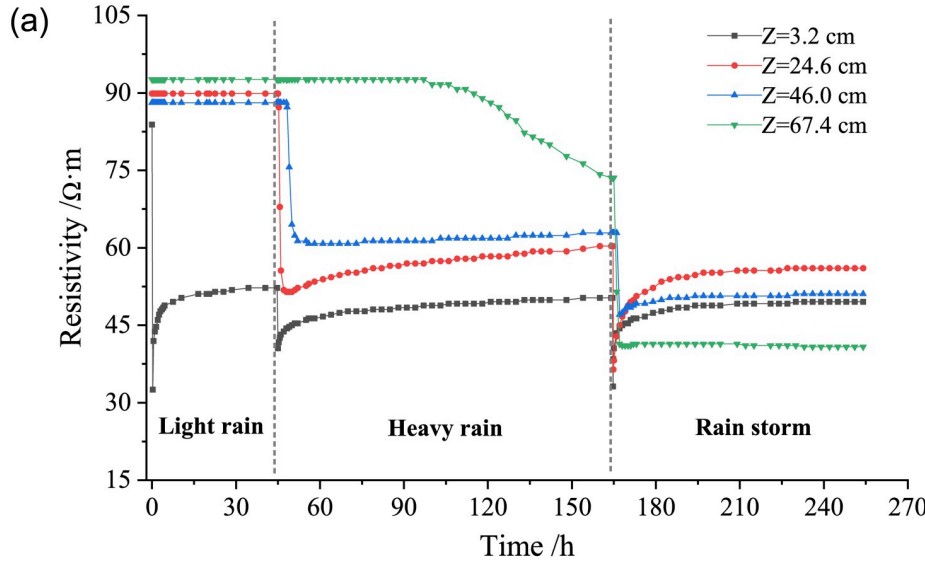

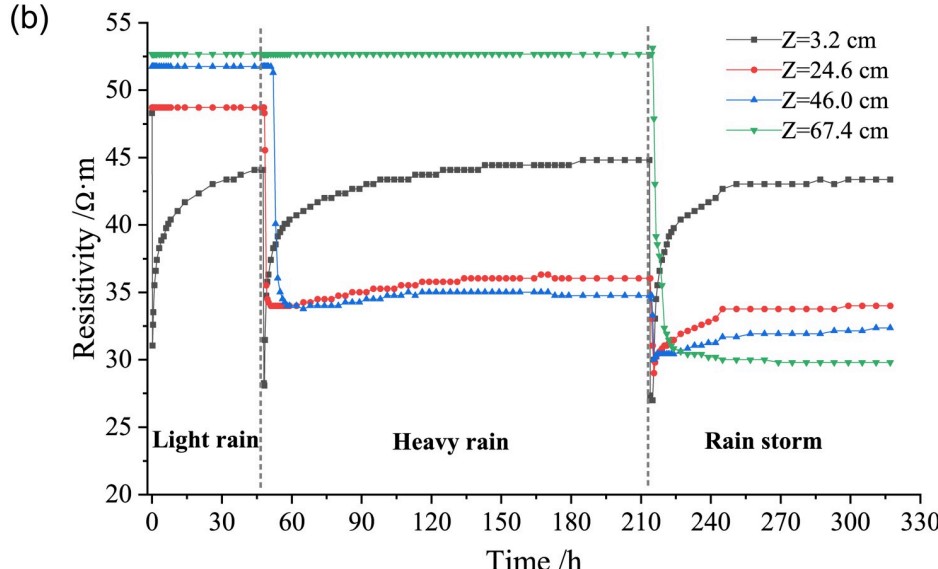

**Fig 5. Variation curves of resistivity of loess with duration under different dry densities.** (a) $\rho_d = 1.42$ g/cm$^3$, (b) $\rho_d = 1.58$ g/cm$^3$.

tends to reach the basic equilibrium state, the resistivity of loess with dry density of 1.42 g/cm$^3$ decreases from 88.1 Ω·m to 62.3 Ω·m by 29.3%. And the resistivity of loess with dry density of 1.58 g/cm$^3$ decreases from 51.8 Ω·m to 30.0 Ω·m by 42%, which is higher than that of loess with dry density of 1.42 g/cm$^3$, it indicates that the greater the dry density, the greater the impact on the resistivity in the process of water migration. The fourth layer (Z = 67.4 cm) of the loess column is at a large depth, and it takes a certain time for the water to migrate to this depth. Therefore, the resistivity at this depth does not change with the continuous test time. When the water migrates to that depth, the resistivity decreases. In the rainstorm stage, the resistivity decreases sharply. With the migration of water to the deeper layer of soil, the resistivity at this depth gradually tended to be basically stable.

## Water migration characteristics in compacted loess under rainfall infiltration

**Water migration characteristics in compacted loess under light rain.** In order to study the characteristics of water migration in compacted loess, the loess columns with different dry densities ($\rho_d$ = 1.42g/cm$^3$, 1.50g/cm$^3$, and 1.58g/cm$^3$) were studied under light rain, the volumetric water content and resistivity at the depth of each layer in the loess column were analyzed at the initial time of the test, the light rain duration of 5 hours, 10 hours, 15 hours and end of light rain, so as to reveal the water migration law at different depths of the loess column under light rain. Figs 6 and 7 respectively show the variation curves of volumetric water content and resistivity with the depth in loess columns with different dry densities under light rain.

From the initial time of rainfall to the basic stability of water migration, the depth of water migration in the loess columns with different dry densities does not exceed the second layer, that is, the depth of water migration is less than 24.6 cm. At the first layer of the loess column (Z = 3.2 cm), with the continuous rainfall infiltration time, the volumetric water content of

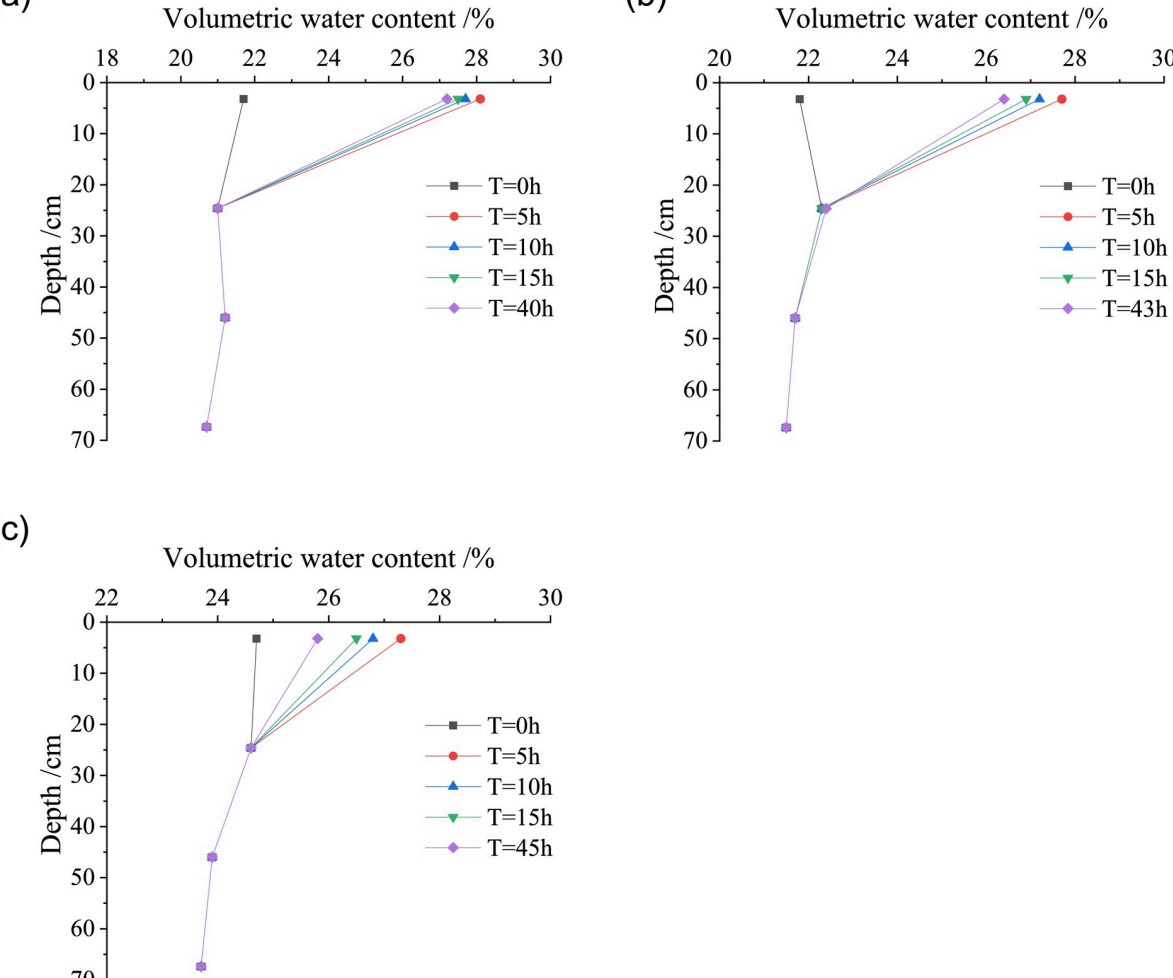

**Fig 6. Variation curves of volumetric water content with the depth of the loess column under light rain.** (a) $\rho_d$ = 1.42g/cm$^3$, (b) $\rho_d$ = 1.50g/cm$^3$, (c) $\rho_d$ = 1.58g/cm$^3$.

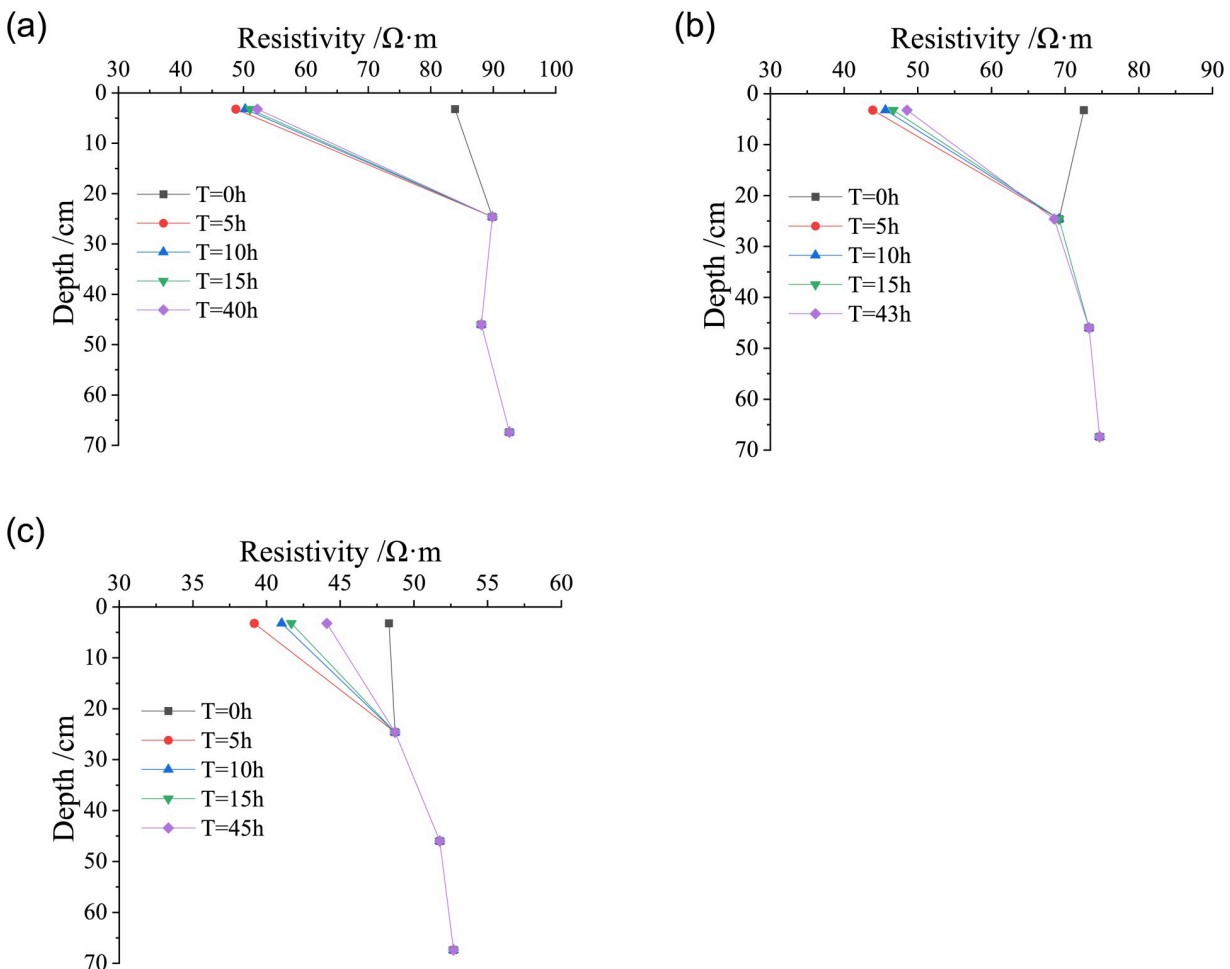

**Fig 7. Variation curves of resistivity with the depth of the loess column under light rain.** (a) $\rho_d = 1.42g/cm^3$, (b) $\rho_d = 1.50g/cm^3$, (c) $\rho_d = 1.58g/cm^3$.

the loess column first increases and then decreases, and the resistivity shows the law of first decreasing and then increasing. When the dry density of the loess column is small, the porosity of the sample is relatively large, and water infiltration rate is faster in the case of the same amount of water. The upper pore is filled with water slowly, the pore water pressure becomes larger and the matrix suction becomes smaller, which leads to the suction of the lower layer loess being greater than that of the upper layer [27]. Under the action of the matrix suction of the lower layer loess and the gravity of water, the water infiltrates into the lower layer, resulting in the gradual increase of the volumetric water content and a reduction in resistivity of the lower layer loess.

With the increasing depth of rainfall infiltration, the volumetric water content decreases and the resistivity increases, and finally both of them are basically stable. The main reason is that the rainfall intensity is far less than the saturated permeability coefficient in the light rain stage, and the water is completely infiltrated. With the increasing depth of rainfall infiltration, the volumetric water content of loess decreases gradually and the resistivity increases, and finally reaches basic stability. The main reason is that the rainfall intensity is far less than the saturated permeability coefficient under light rain, and the water is completely infiltrated. In the light rain stage, with the increase of dry density of the loess column, due to the limited

depth of rain infiltration, the distribution of rainfall infiltration line in the loess column shows a relationship of "Y" shape with the initial state.

**Water migration characteristics in compacted loess under heavy rain.** The volumetric water content and resistivity at the depth of each layer in the loess column were analyzed at the initial time of the test, the heavy rain duration of 10 hours, 30 hours, 50 hours and end of heavy rain, in order to study the water migration characteristics at different depths of the loess column with different dry densities ($\rho_d = 1.42$ g/cm$^3$, 1.50 g/cm$^3$, and 1.58 g/cm$^3$) under heavy rain. Figs 8 and 9 respectively present the variation curves of volumetric water content and resistivity with the depth of the loess columns with different dry densities under heavy rain.

It can be observed from Figs 8 and 9 that in the process of rainfall infiltration from the initial time of heavy rain to the basic stability, due to the different compaction degree of the loess column, the infiltration depth of rainfall gradually decreases with the increase of dry density of the loess column. When the dry densities of soil column are 1.42 g/cm$^3$ and 1.50 g/cm$^3$, the depth of water migration reaches at the fourth layer of the soil column (Z = 67.4 cm), and there is water seepage at the bottom drain of the model test device, indicating that the water

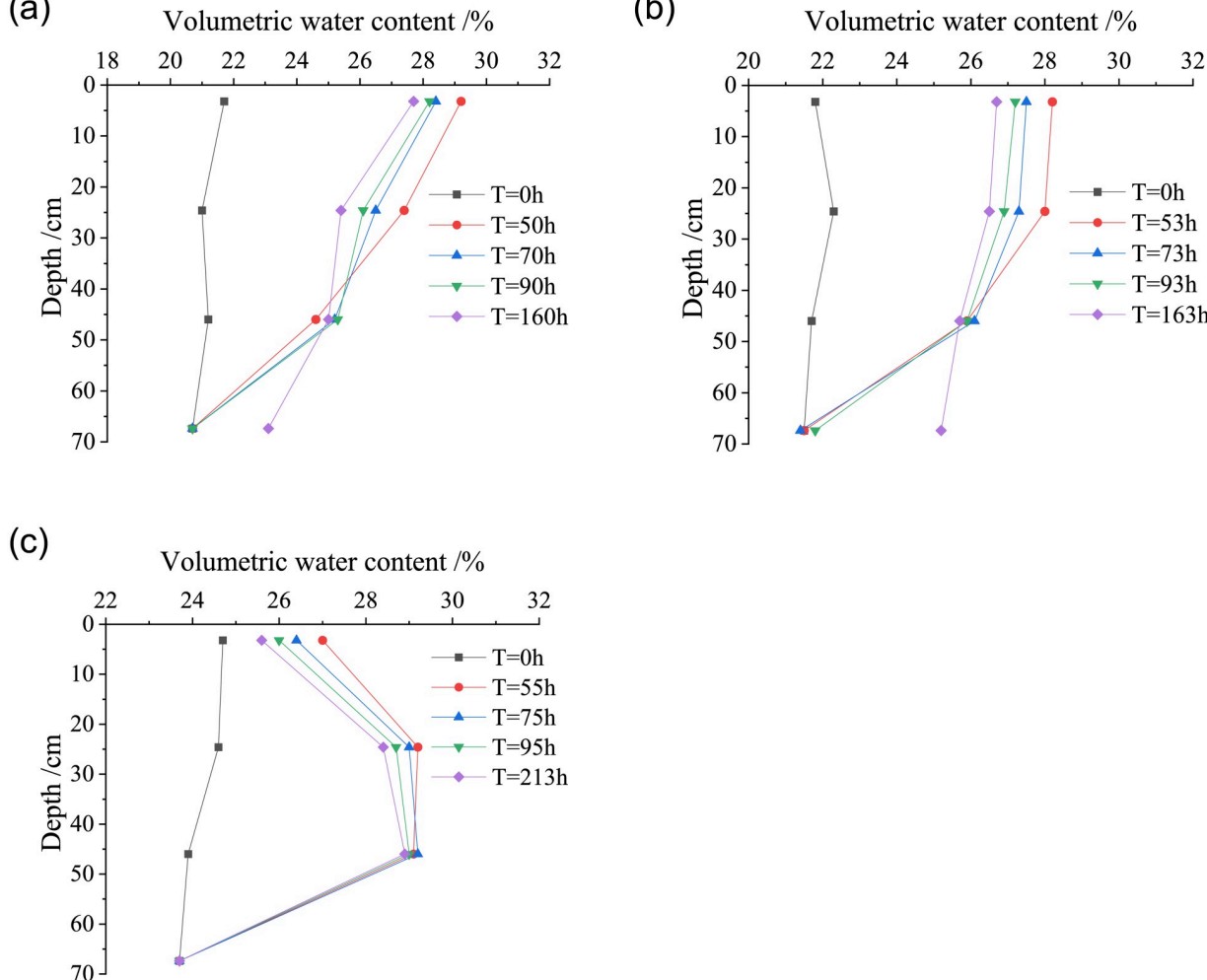

**Fig 8. Variation curves of volumetric water content with the depth of the loess column under heavy rain.** (a) $\rho_d$ = 1.42g/cm$^3$, (b) $\rho_d$ = 1.50g/cm$^3$, (c) $\rho_d$ = 1.58g/cm$^3$.

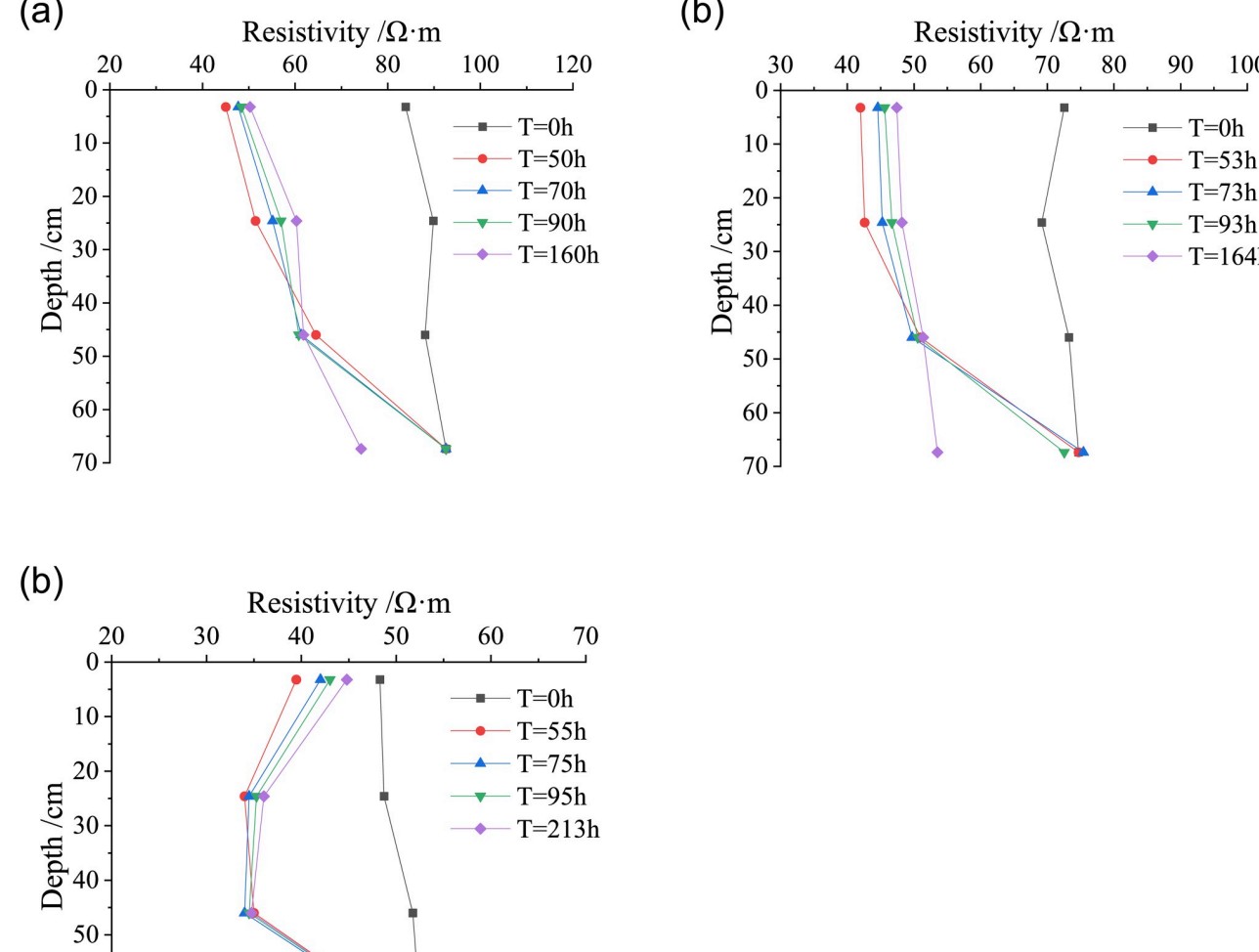

**Fig 9. Variation curves of resistivity with the depth of the loess column under heavy rain.** (a) $\rho_d$ = 1.42g/cm$^3$, (b) $\rho_d$ = 1.50g/cm$^3$, (c) $\rho_d$ = 1.58g/cm$^3$.

migration depth has exceeded 70 cm. While the depth of water migration in the loess column with dry density of 1.58 g/cm$^3$ is located between the third layer and the fourth layer of the soil column (56.7 cm <Z< 67.4 cm). With the increase of dry density of the loess column, volumetric water content and resistivity of loess to achieve the basic stability of the time required becomes longer, and the change depth becomes shallower.

In the range of 0-30cm depth of the loess column, with the continuous rainfall time, the volumetric water content of loess gradually decreases, while the resistivity table shows a gradual changing trend, which indicates that the surface water has been infiltrating downward. At the same rainfall time, the volumetric water content decreases with the increase of infiltration depth in the loess column of dry density ($\rho_d$ = 1.42 g/cm$^3$, 1.50 g/cm$^3$), while the volumetric water content increases firstly and then decreases with the increase of infiltration depth in the loess column of dry density ($\rho_d$ = 1.58 g/cm$^3$), and the resistivity of loess shows the opposite rule.

During the downward infiltration of water in the loess column, the volumetric water content of loess first increases significantly. With the continuous rainfall time, the volumetric water content of loess decreases gradually until it reaches a basically stable state. With the increase of the dry density of the loess column, the rainfall infiltration line in the loess column gradually presents a relationship of "D" shape distribution with the initial state under heavy rain.

**Water migration characteristics in compacted loess under rainstorm.** The volumetric water content and resistivity of loess at the depth of each layer in the loess column were selected at the initial time of the test, the rainstorm duration of 10 hours, 30 hours, 50 hours and end of rainstorm, respectively, so as to analyze the variation law of volumetric water content and resistivity at different depths of each layer in the loess column under the rainstorm. As shown in Figs 10 and 11, the curves of volumetric water content and resistivity versus depth of the loess columns with different dry densities under rainstorm.

From the beginning of rainstorm to the water migration reaching the basic stability, there is water seepage at the bottom drain of the model test device in each loess column with different

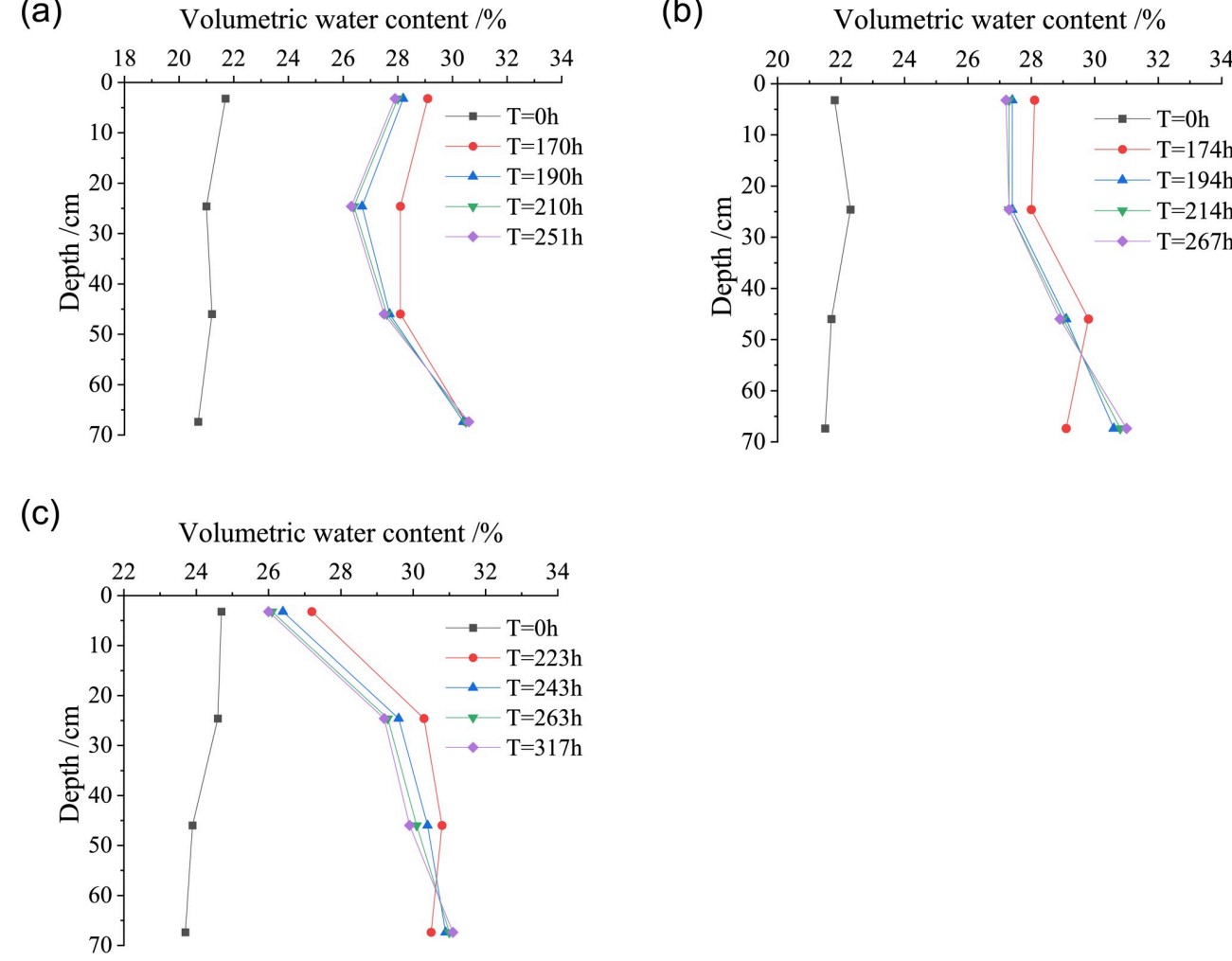

**Fig 10. Variation curves of volumetric water content with the depth of the loess column under heavy rain.** (a) $\rho_d$ = 1.42g/cm$^3$, (b) $\rho_d$ = 1.50g/cm$^3$, (c) $\rho_d$ = 1.58g/cm$^3$.

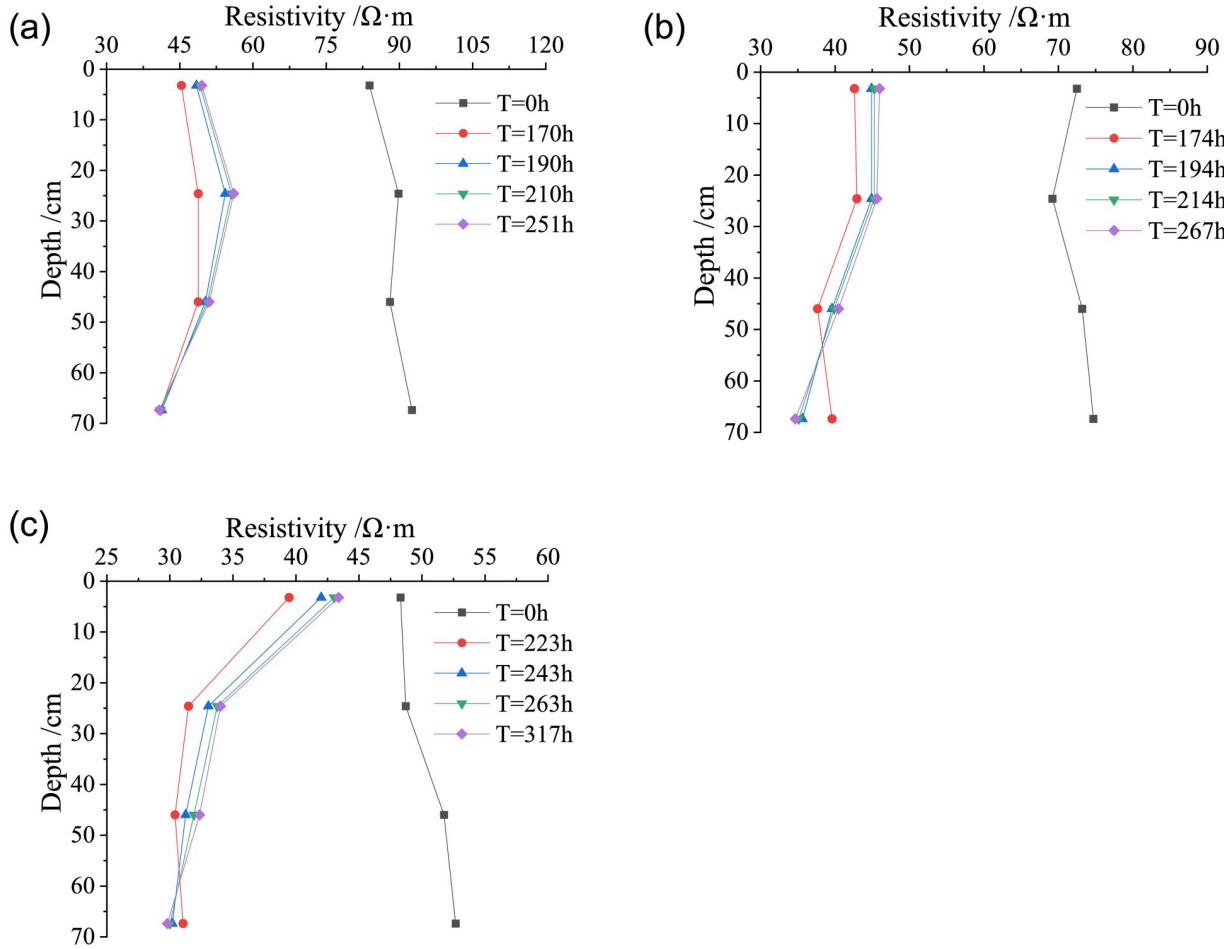

**Fig 11. Variation curves of resistivity with the depth of the loess column under heavy rain.** (a) $\rho_d$ = 1.42g/cm$^3$, (b) $\rho_d$ = 1.50g/cm$^3$, (c) $\rho_d$ = 1.58g/cm$^3$.

dry densities, and the water migration reaches the maximum depth of the loess column ($Z_{max}$ > 70 cm). The change of water content in shallow loess is sensitive under different rainfall conditions [28]. In the range of 0-50cm depth of the loess column, with the continuous rainfall time, the volumetric water content of loess decreases gradually, while the resistivity shows a trend of gradual increase, which reveals that the water component has been infiltrating deeper into the loess column under the action of rainstorm. When the water migration is close to the bottom of the loess column, due to the gradual accumulation of water with the duration of rainfall, the volumetric water content of loess gradually increases and the resistivity decreases gradually.

At the same rainfall time, when the dry densities of the soil column are 1.42 g/cm$^3$ and 1.50 g/cm$^3$, the volumetric water content of loess firstly decreased and then increased with the increasing depth of rainfall infiltration, while the volumetric water content of loess increases with the increasing depth of rainfall infiltration at $\rho_d$ = 1.58 g/cm$^3$, and the resistivity shows the opposite trend. With the continuous rainfall time, a large amount of water infiltrates downward in the loess column. With the increase of dry density of the loess column, the volumetric water content at the depth of each layer in the loess column increases gradually, the

rainfall infiltration line in the loess column gradually shows a relationship of "Λ" shape distribution with the initial state under rainstorm.

## Discussion

### Relationship of volumetric water content-saturation-resistivity

The resistivity ($\rho$) is related to volumetric water content ($\theta_v$) and dry density ($\rho_d$), if the applied soil has the same properties (referring to the same granulometric composition, same ambient temperature, and the same chemical composition of pore water, etc. [29, 30]. To study the relationship of the resistivity ($\rho$) with $\theta_v$ and $\rho_d$, the $\theta_v$ and $\rho_d$ can be normalized by saturation ($S_r$) of loess, the derivation is as follows:

$$S_r = \frac{wG_s}{e_0} \tag{1}$$

Because

$$e_0 = \frac{G_s}{\rho_d} - 1 \tag{2}$$

$$\theta_v = \frac{w\rho_0}{(1+w)\rho_w} = \frac{w\rho_d}{\rho_w} \tag{3}$$

From Eqs (1), (2) and (3), which can get $S_r$:

$$S_r = \frac{\theta_v G_s}{G_s - \rho_d} \tag{4}$$

Where the $\rho_0$ is the density of the soil (g/cm$^3$); $\rho_w$ is the density of water (g/cm$^3$), which is 1g/cm$^3$ at room temperature; $w$ is water content (%); $e_0$ is the initial void ratio; $G_s$ is the specific gravity of soil particle.

Based on the above test results, the relationship between $\rho$ and $S_r$ can be plotted as a curve shown in Fig 12, which can be fitted to obtain the following equation:

$$\rho = 16.299 S_r^{-2.096} \tag{5}$$

or

$$\rho = 16.299 \left(\frac{\theta_v G_s}{G_s - \rho_d}\right)^{-2.096} \tag{6}$$

Fig 12 illustrates that $\rho$ decreases with $S_r$. Specifically, the $\rho$ increases rapidly with $S_r$ in the range of low $S_r$. The increase rate of $\rho$ with $S_r$ gets smaller after $S_r$ exceeds a particular value and $\rho$ tends to be stable at a certain value at very low $S_r$, which is consistent with the results reported by the references [31, 32]. The $S_r$ normally presents a positive relationship with dry density and water content, the larger the $S_r$ is, the more dynamic the cation ions in the pore water within the particle boundary [33, 34]. Additionally, for the loess of the same properties, the better conductivity means the lower resistivity. Therefore, it is reasonable to probe the water migration by monitoring the evolution of the resistivity in the soil. In the loess high-filling practical engineering, the change of resistivity can be monitored by embedding resistivity sensors at different depths of the site, and then the purpose of predicting the change law of soil saturation or water field can be achieved from the evolution process of resistivity at different

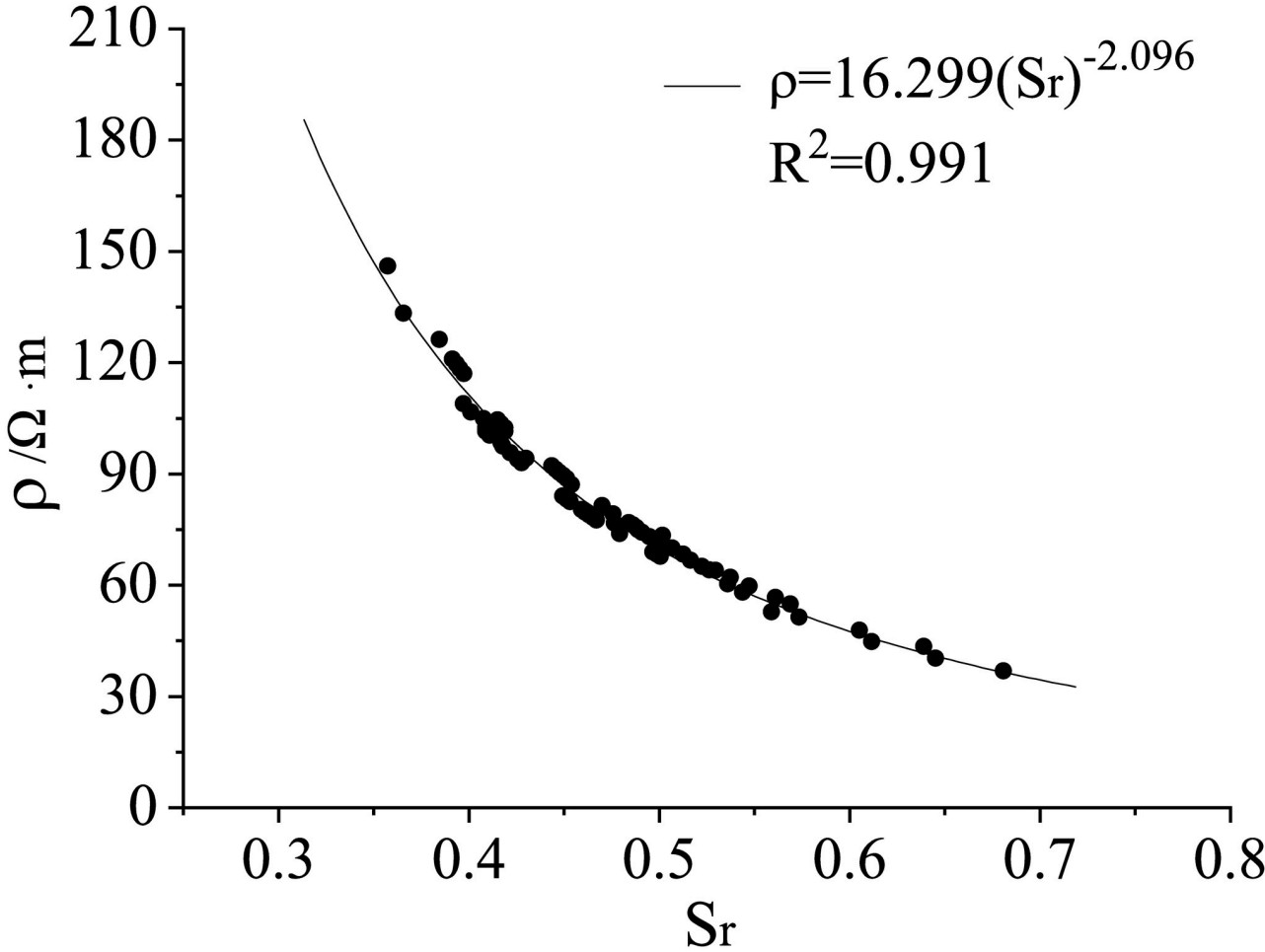

**Fig 12. The fitting curve of resistivity and saturation.**

depths of the site, which is of great value for studying the change law and deformation mechanism of water field in the site of loess high-filling engineering.

### The systematic discussion on the characteristics of water migration in compacted loess under different rainfall infiltration conditions

To investigate the effect of the different rainfall conditions on the water migration, selecting four states of the loess column with the same dry density: initial state, state after light rain, state after heavy rain, and state after rainstorm. Subsequently, the volumetric water content and resistivity was analyzed at different depths in the loess column to obtain the relationship of both volumetric water content and resistivity with the depth of soil column, unveiling the change of the infiltration line under different rainfall.

From Fig 13, under the condition of light rain, as the daily rainfall is only 7.1 mm, the maximum migration depth of water is limited to less than 24.6 mm for the soil of different dry density when the water migration becomes stable in the loess column. Therefore, in this situation, the volumetric water content shows no change with the depth of the loess column within the depth region where the water migration has not reached, meaning that the rainfall infiltration line shows a relationship of "Y" shape distribution with the initial state. In the heavy rain, the

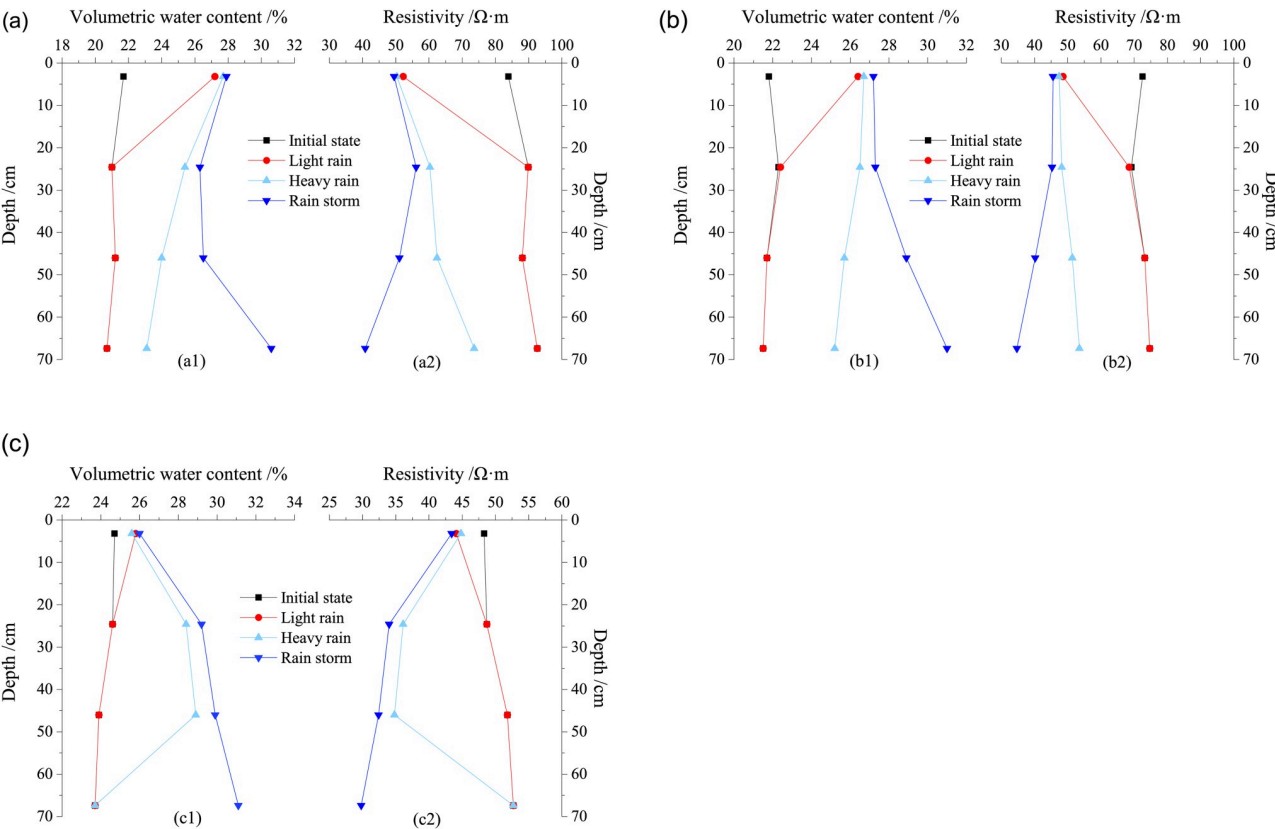

**Fig 13. Relationship of both volumetric water content and resistivity with the depth of the loess column under different rainfall.** (a) $\rho_d = 1.42g/cm^3$, (b) $\rho_d = 1.50g/cm^3$, (c) $\rho_d = 1.58g/cm^3$.

daily rainfall of heavy rain (35.4 mm) is 5 times that of light rain, thus, the volumetric water content initially increases, subsequently, decreases with the depth of the loess column owing to the water amount when the rainfall infiltration becomes stable in the soil of different dry density, indicating the rainfall infiltration line shows a relationship of "D" shape distribution with the initial state when the dry density of loess increases. In the case of rainstorm, the maximum depth that the water migration can reach is more than 70 cm, a large amount of water migrates to the low part of the loess column, the volumetric water content gradually increases until stability is maintained through the whole soil column. The water constantly migrates to the lower depth of the loess column, thus, gathered at the bottom of the loess column, leading to a continuing increase of the volumetric water content. In this scenario, "Λ" shape distribution is attained between the rainfall infiltration line and the initial state when the dry density of loess increases, while the resistivity shows an opposite trend to the volumetric water content.

The previous study on the water migration in compacted loess also revealed some properties of the water migration. In this study, the detailed shapes of the rainfall infiltration line are exposed in the loess under the condition of light rain, heavy rain, and rainstorm. The literatures [35, 36] reported that the water migrates by unsaturated seepage or vaporization in loess, which was concluded by studying the behavior of water migration in the loess in the natural rainfall generated by artificial precipitates. Under different rainfall conditions, the volumetric water content increases with the depth of water migration in loess, its change is gradually stagnant, and the increasing rate of the water content decreases. Water migration showed an ellipse shape towards the depth of original loess, as reported in literature [37], which was

different from the results in this study. This study disclosed the behavior of the water migration by analyzing the volumetric water content and resistivity of remolded loess in the laboratory model test. The water migration demonstrated a unique shape against the depth of the loess column under different rainfall conditions. In addition, the reason that the rainfall infiltration line shows a relationship of "Λ" shape distribution with the initial state is because the height of the loess column is not enough to reflect the whole part of the water migration. The previous study on the water migration in the loess indicated that the volumetric water content in the shallow part of the soil is sensitive to the intensity of rainfall under rainfall conditions. The depth of the water migration is normally limited to 2 m in general or a maximum of 3 m in depth [19, 38–40]. In following model tests of water migration in the loess column, the loess column will be extended to 2–3 m in order to disclose the features of the water migration in loess under more realistic conditions of rainfall. Through real-time understanding of the variation law of water migration in the practical engineering site, the variation law of water field can be quickly obtained, and the deformation and stability of deep loess high-filling engineering or high slope engineering can be monitored for long-term, which is convenient for safety early warning and disaster prevention.

## Conclusions

1. This study developed a model test device for testing water migration in compacted loess, and the volumetric water content and conductivity characteristics of loess are introduced into this test device, and more deeply disclosed the features of the water migration in compacted loess, achieving a good understanding on the behavior of the water migration from the delicate perspective of the volumetric water content and the resistivity of loess, which can provide a new methodology for the study of the stability of loess slope and the water migration characteristics in loess foundation of high-filling engineering under rainfall infiltration.

2. By employing the developed model test device of water migration characteristics and conducting the study on the water migration in compacted loess under different rainfall conditions, the evolution pattern of the volumetric water content and resistivity at the different depth of the loess column is achieved. The resistivity decreases with the increase of the volumetric water content at the same depth of the loess column, in this way, the features of the water migration can also be reflected in the change of the resistivity.

3. There is an intimate relationship between $\rho$ and $\theta_v$, $\rho_d$ of loess, after normalizing $\theta_v$ and $\rho_d$, arriving the equation of $S_r$ against $\rho$ by fitting the curve of the numerical data. The resistivity of the loess gradually decreases with the saturation, and it varies greatly with the saturation in the low-saturation range. The amplitude of variation decreases with the continuous increase of the saturation.

4. Under different rainfall intensities, the characteristics of rainfall infiltration at different depths of the loess column show a particular pattern: with the increasing of dry density of the loess column, the rainfall infiltration line presents a relationship of "Y", "D" and "Λ" shape distribution, respectively, under the condition of light rain, heavy rain and rainstorm.

## Author Contributions

**Formal analysis:** Shibin Zhang.

**Investigation:** Yani Lu, Chengzhi Huang.

**Methodology:** Shibin Zhang, Tielin Han, Yani Lu.

**Resources:** Tielin Han, Yani Lu.

**Writing – original draft:** Shibin Zhang.

**Writing – review & editing:** Shibin Zhang, Tielin Han, Chengzhi Huang, Peng Zhao.

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
