## [Decision Letter · Decision Letter 0]

28 Jun 2022

PONE-D-22-12253Experimental Study of Water Migration Characteristics in Compacted Loess Subjected to Rainfall InfiltrationPLOS ONE

Dear Dr. Han,

Thank you for submitting your manuscript to PLOS ONE. After careful consideration, we feel that it has merit but does not fully meet PLOS ONE’s publication criteria as it currently stands. Therefore, we invite you to submit a revised version of the manuscript that addresses the points raised during the review process.

We look forward to receiving your revised manuscript.

Kind regards,

Chun Liu

Academic Editor

PLOS ONE

Journal Requirements:

“This research was funded by the National Natural Science Foundation of China (Grant No. 41907259) (Yani Lu)and Natural Science Foundation of Shaanxi Provincial of China (Grant No. 2021JQ-463)(Tielin Han).YES -Specify the role(s) played.”

Reviewers' comments:

Reviewer's Responses to Questions

**Comments to the Author**

1. Is the manuscript technically sound, and do the data support the conclusions?

Reviewer #1: Yes

2. Has the statistical analysis been performed appropriately and rigorously? 

Reviewer #1: Yes

3. Have the authors made all data underlying the findings in their manuscript fully available?

Reviewer #1: Yes

4. Is the manuscript presented in an intelligible fashion and written in standard English?

Reviewer #1: Yes

5. Review Comments to the Author

Reviewer #1: In this paper, the water migration characteristics of compacted loess under different rainfall conditions were studied, and the corresponding results were obtained. However, there are still some confusions about the article, as follows. I recommend that the article should be published after a substantial revision.

1. Introduction: Authors cite some papers here for literature review. However, most of the text is a brief introduction to these studies and the generalization and summary of these studies are relatively lacking. In this case, the necessity of the article is unconvincing. Authors are advised to make a substantial revision of the Introduction, systematically review of previous research and highlight the innovation are necessary.

2. Line 71: ‘were’ not ‘was’

3. Fig. 1: Figure b should also add a corresponding legend description.

4. Fig. 2: ‘processes’ not ‘process’

5. Line 149: The latitude and longitude coordinates of the sampling point need to be given.

6. Line 155: Please explain the reasons for these three dry density choices.

7. Eq. 5 and 6: The format is different from other formulas

8. Authors shows a lot of results in the results section, but some phenomena are not further explained. E.g., line 223-229: No further analysis on conductivity changes is given here.

9. As the authors say, understanding the water transport properties of loess has important practical significance. In the discussion, however, I did not find the authors to make any practical recommendations based on the research results.

6. PLOS authors have the option to publish the peer review history of their article (what does this mean?). If published, this will include your full peer review and any attached files.

Reviewer #1: No

---

## [Author Response · Author response to Decision Letter 0]

19 Jul 2022

Summary of amendments

Re: “Experimental Study of Water Migration Characteristics in Compacted Loess Subjected to Rainfall Infiltration”

For PLOS ONE

Ref.: MS No: PONE-D-22-12253

Many thanks for your letter and for the editor and reviewers’ constructive comments concerning our manuscript. We feel that the editor and reviewers’ comments have greatly benefited the manuscript. We have studied comments carefully, and these comments and suggestion have been carefully incorporated into the current manuscript. 

The point-to-point reply for the comments:

Reviewers' comments:

Comment 1. Introduction: Authors cite some papers here for literature review. However, most of the text is a brief introduction to these studies and the generalization and summary of these studies are relatively lacking. In this case, the necessity of the article is unconvincing. Authors are advised to make a substantial revision of the Introduction, systematically review of previous research and highlight the innovation are necessary.

Response: 

Thanks for your kindly suggestion. The section of introduction has been significantly modified in the current manuscript according to the reviewer’s suggestion. Please see

Lines 32-35: By analyzing the stability of slopes under different rainfall intensities, Shimada et al. [9] concluded that the increase of saturation in the slope led to the reduction of matrix suction under rainfall, which had a great impact on the stability of the slope.

Lines 36-38: and concluded that the increase of pore water pressure (the reduction of suction) was mainly concentrated near the slope in the process of rainfall infiltration.

Lines 45-48: Sun et al. [15] studied the influence of rainfall infiltration on the slope stability and the seepage field by using the finite element and limit equilibrium methods, and concluded that rainfall infiltration led to significant changes in the seepage field, especially when the pore water pressure of slope increased greatly, the deep slope failure was prone to occur.

Lines 48-50: The stability analysis mechanism of unsaturated soil slope is complicated and greatly affected by various factors under rainfall infiltration.

Lines 58-59: and concluded that the safety factor of slope was changing during rainfall infiltration.

Lines 59-60: The wetting depth of slope is an important index to evaluate the slope instability induced by rainfall.

Lines 63-64: It was necessary to characterize the field infiltration and wet front movement induced by natural rainfall.

Lines 70-72: and the results showed that the water infiltration depth and infiltration rate increase with the increase of rainfall intensity during rainfall.

Comment 2. Line 71: ‘were’ not ‘was’. 

Response: 

Thanks for your kindly suggestion. According to the reviewer’s comments, the relevant content has been revised. 

Comment 3. Fig. 1: Figure b should also add a corresponding legend description.

Response: 

Thanks for your kindly suggestion. The authors have made some revision about it according to reviewer’s suggestion, as shown in Fig. 1(b). 

Comment 4. Fig. 2: ‘processes’ not ‘process’.

Response: 

Thanks for your kindly suggestion. The authors have made some revision about it according to reviewer’s suggestion. 

Line 132: Sample preparation processes.

Comment 5. Line 149: The latitude and longitude coordinates of the sampling point need to be given.

Response: 

Thanks for your kindly suggestion. The authors have made some revision about it according to reviewer’s suggestion. 

Lines 158-159: The loess samples were taken from the construction site of loess slope engineering with a latitude and longitude of 36°38′ N, 109°32′ E in New District of Yan'an in China.

Comment 6. Line 155: Please explain the reasons for these three dry density choices.

Response: 

Thanks for your kindly suggestion. Lines 165-170: Three dry densities represent the loess columns with different compaction degree, and the increase of dry density in turn can reflect the gradual compaction of the loess columns. The loess column with dry density of 1.42 g/cm3 is easy to be compacted, while it is difficult to be compacted at dry density of 1.58 g/cm3 in the actual sample preparation process. And the dry density of 1.50 g/cm3 is chosen because it is located in the middle of 1.42 g/cm3 and 1.58 g/cm3.

Comment 7. Eq. 5 and 6: The format is different from other formulas.

Response: 

Thanks for your kindly suggestion. According to the reviewer’s comments, the format of Eq. 5 and 6 have been revised. 

Comment 8. Authors shows a lot of results in the results section, but some phenomena are not further explained. E.g., line 223-229: No further analysis on conductivity changes is given here.

Response: 

Thanks for your kindly suggestion. According to the reviewer’s comments, the relevant contents have been added in the manuscript. Please see

Lines 226-231: When the dry density of the loess column is small, its compaction degree is low, and the porosity of soil is relatively large. Under the same amount of rainfall, the rainfall infiltration rate is faster. With the increase of dry density of loess column, the soil is gradually compacted and the porosity of soil decreases. The water infiltration rate decreases and the water transport capacity is poor. Therefore, the time required for the rainfall infiltration to reach the basic stable state increases significantly during water migration in the loess column.

Lines 246-251: At the initial moment of rainfall, the amount of water in the surface layer of the loess column increases sharply, and it is too late for the water to infiltrate into the deep layer of the loess column. The saturation of surface soil rapidly increases, which makes the resistivity greatly decrease. As the rain infiltrates into the deep layer of the loess column, the saturation of the surface soil gradually decreases until it reaches an equilibrium state, while the saturation of the deep soil gradually increases, and the resistivity of soil changes accordingly.

Lines 253-256: This also reflects that as the rain infiltrates into the deep layer of the loess column, the water volume between soil pores increases rapidly at first and then decreases gradually until the water no longer migrates to the deeper layer, reaching a basic equilibrium state.

Comment 9. As the authors say, understanding the water transport properties of loess has important practical significance. In the discussion, however, I did not find the authors to make any practical recommendations based on the research results.

Response: 

Thanks for your kindly suggestion. The authors have made some revision about it according to reviewer’s suggestion. 

Lines 393-397: In the loess high-filling practical engineering, the change of resistivity can be monitored by embedding resistivity sensors at different depths of the site, and then the purpose of predicting the change law of soil saturation or water field can be achieved from the evolution process of resistivity at different depths of the site, which is of great value for studying the change law and deformation mechanism of water field in the site of loess high-filling engineering.

Lines 452-455: Through real-time understanding of the variation law of water migration in the practical engineering site, the variation law of water field can be quickly obtained, and the deformation and stability of deep loess high-filling engineering or high slope engineering can be monitored for long-term, which is convenient for safety early warning and disaster prevention.

---

## [Editor Report · Decision Letter 1]

25 Aug 2022

Experimental Study of Water Migration Characteristics in Compacted Loess Subjected to Rainfall Infiltration

PONE-D-22-12253R1

Dear Dr. Han,

We’re pleased to inform you that your manuscript has been judged scientifically suitable for publication and will be formally accepted for publication once it meets all outstanding technical requirements.

Kind regards,

Chun Liu

Academic Editor

PLOS ONE
---

## [Editor Report · Acceptance letter]

30 Aug 2022

PONE-D-22-12253R1 

Experimental Study of Water Migration Characteristics in Compacted Loess Subjected to Rainfall Infiltration 

Dear Dr. Han:

I'm pleased to inform you that your manuscript has been deemed suitable for publication in PLOS ONE. Congratulations! Your manuscript is now with our production department. 

Kind regards, 

on behalf of

Dr. Chun Liu 

Academic Editor

PLOS ONE